# New Blended Learning Enriched after the COVID-19 Experience? Students' Opinions

**Ivana Simonova \*, Ludmila Faltynkova and Katerina Kostolanyova**

Faculty of Education, University of Ostrava, 701 03 Ostrava, Czech Republic
\* Correspondence: ivana.simonova@osu.cz

**Abstract:** Online distance instruction (ODI), as a response to COVID-19 restrictions, had a great impact on education around the world. The main objective of the presented research is to monitor students' opinions and answer the question of whether teachers enriched face-to-face lessons enhanced by digital technologies (i.e., blended learning) after the COVID-19 pandemic with the methods and tools that they used during ODI in the pandemic. Data were collected at selected upper secondary and higher education institutions for medical staff in the Czech Republic (N = 488) using online questionnaires in autumn 2021 (end of ODI) and in autumn 2022 (a year of blended learning). The questionnaires consisted of 35 items that required students' opinions on the Likert scale, multiple-choice items with one or more answers, and open answers. The frequency of occurrence was monitored according to four criteria: (1) First contact and teacher-student communication, (2) learning content acquisition, (3) learning content delivery and assessment, (4) students' final feedback on ODI. The results did not show much enrichment of blended learning using the experience from ODI. On the one hand, presentations were more frequently exploited in blended learning than in ODI. On the other hand, teachers did not use one channel to deliver study materials and conduct communication as they had done during ODI. We cannot deduce the reasons for teachers' behavior from the collected data, but students were sure that these changes did not help them in learning.

**Keywords:** online distance; blended learning; upper secondary; higher education; questionnaire; medical staff; Czech Republic

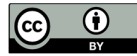

## 1. Introduction

Similarly to most countries around the world, in the Czech Republic, the COVID-19 pandemic suddenly disrupted face-to-face instruction within which blended learning was widely applied. Schools were closed by law to protect public health, and instead, emergency remote teaching, and online distance instruction were conducted. In the context of this article, (a) face-to-face (f2f) instruction refers to instruction that occurs in a physical classroom, the teacher and the student are present in a room, and the teacher is providing or facilitating instruction [1], (b) blended learning (BL) is understood as a combination of f2f lessons and online distance learning, that is, autonomous a/synchronous learning enhanced by digital technologies conducted individually or in pairs, teams from home. Depending on the habit and context, the term learning also stands for blended teaching or instruction as well (authors' definition), (c) emergency remote teaching (ERT), in contrast to online distance instruction (ODI), which is planned and designed to be conducted online [2], is a temporary way in which lessons can be conducted due to a crisis. The main purpose is to bridge the problematic period and then to return to the original manner of teaching [3], (d) online distance instruction applies to an educational process where teachers and learners are separated by distance. Learners are taught in online classes using digital technology. Through the shared screen, they can attend lectures and discuss the

topics, do exercises and tests, practice, and assess new knowledge without having to be present in a classroom [4].

*1.1. The Course of Teaching and Learning under COVID-19 Restrictions in the Czech Republic*

Following the development of COVID-19 pandemic in the Czech Republic, time of instruction under pandemic restrictions can be structured into three main periods. Based on their description, we set the main objective of the research.

(1) Half of March 2020–June 2020: The first period carries typical features of emergency remote teaching (ERT). It started suddenly, unexpectedly and was not expected to last more than two-three weeks. From July to August 2020, the summer holidays were held, when all schools were closed. During the holidays, the Czech government and the Ministry of Education prepared some acts and directions. The most important of them was that attendance was compulsory [5,6]. To meet this requirement, it was necessary to equip schools with hardware that learners could borrow. The Ministry also published a set of didactic recommendations on how to teach online. However, the document was too theoretical. In the end, schools held training courses for teachers themselves or with the help of regional didactic centers.

(2) Half of September 2020–January 2021: The second period was characterized as online distance instruction (ODI). At the beginning of September, teachers had a short period of time to teach learners how to learn online distance. Then, the schools were closed again. During the summer months of 2020, schools were equipped with notebooks, learners from socially weak families could borrow them. However, the problems of the cost of Internet access and technical problems related to the low quality of connection were still alive. Moreover, some lazy learners found out how to pretend to have technical problems and cheat in online testing, so teachers had to face new problems they had not known before. This period finished at the end of January 2021, which was the end of the first half of the school year at upper secondary schools and the end of the exam period at higher education institutions.

(3) February 2021–June 2021: The third period, also defined as ODI, when lessons were held online without a break for next five months. Teachers and learners were tired, demotivated, stressed by demanding preparation of ODI, conducting lessons, doing homework and providing feedback, online communication, solving technical problems, a lack of social contacts, and many others. In addition, the students at both schools for medical staff were tired of long hours they spent in hospitals having voluntary practice, the whole society was stressed by high amounts of COVID-19-infected patients, etc. During summer holidays in July–August 2021, new directions, more helpful than a year before, were published by Czech School Inspectorate [7,8]. The COVID-19 statistics were improving step-by-step, and since September 2021, only f2f lessons have been taught. ODI is acceptable as an approach within higher education, mainly in large groups of students or other appropriate situations. In upper secondary schools, strong preference is given to f2f lessons. Practical lessons to train medical staff and higher medical staff are solely held f2f.

(4) Since September 2021: Post-COVID period began. Thus, after 18 months of teaching and learning under the COVID-19 pandemic restrictions, when ERT or ODI were conducted, f2f instruction re-started again, supported by digital technologies, that is, blended learning re-appeared. During the period of restrictions, teachers and learners were forced to improve their knowledge and skills in using digital technologies for educational purposes, and they could use new methods and tools that they did not know before. The question is whether they really did so in the new, post-COVID blended learning. In other words, did teachers enrich blended learning in the post-COVID period by reflecting on the experience from online distance instruction? Answering this question from the students' view is the main objective of the article.

### 1.2. Literature Review

In accordance with our research and to obtain a rather complex view, we intended to structure the observed span of education into the periods that followed the development in the Czech Republic. However, the usage of the structure did not show to be useful for the review. As we can see from the review, the periods the COVID-19 restrictions were applied for were different in various countries, but the worldwide approach to this topic is necessary. Therefore, articles included in this review were selected by applying the keywords COVID-19 and online distance, COVID-19 and emergency remote teaching, post-COVID AND blended learning in recognized databases Web of Science and Scopus within 2020–2023. Intentionally, no keywords related to medical staff or students were used so that the sample was not limited before considering the subject. After doing so, the articles on medical staff education were a matter of priority, however, there were not many.

According to the definition, emergency remote teaching should be conducted for a short period of time necessary for restoring the original or new but stable conditions. However, as we can see in documents published by OECD [9], Czech School Inspectorate [7], the Ministry of Education [10], the terms ERT and ODI are often used interchangeably, so we will not distinguish the terms in this review, as other scholars do not do it either.

We called the manner of teaching and learning ODI during the whole period of COVID-19, but in reality, it was ERT, as Rohlikova (in Neumajer, n. d.) emphasizes [11]. According to Zlamalova [2], distance education is mainly based on autonomous learning, when study materials are designed with a special structure that helps simulate teacher-learner f2f didactic communication. Furthermore, if digital technologies enhance learning, learners must receive a manual on how to study online distance and continuous motivational support. Learning objectives should be expressed by active verbs following Bloom's taxonomy of educational objectives [12]. One idea should be expressed in a separate paragraph, and several paragraphs should form a chapter. Wide margins should be left empty for making notes. Short questions should be included after subchapters, questions requiring deeper thinking or teamwork after the chapter. Answers to questions can be provided in a key or discussed in f2f or f2screen sessions. Together with exercises and tests, they provide feedback to learners and assess their knowledge. In online or f2f discussions, teachers and learners can share their experiences and samples of good practice. They support motivation to learn and target future needs. The list of terminology and recommended sources for study should be appended.

Cramarenco et al. [13] conducted a review of surveys on student perceptions of online education between March 2020 and September 2022. Using the keyword emergency remote teaching, they searched in databases Scopus and Web of Science and found 154 articles addressing mostly positive, negative, or neutral student opinions. Their analysis revealed that the field had not been sufficiently explored. They recommend that universities must prepare more consciously for embedding current technological challenges to cope with unforeseen situations, such as an immediate switch from the classic f2f teaching to online based on digital technology education. This is rather a general, wide, and frequently presented recommendation. Unfortunately, the study does not point out a concrete way to meet the requirements, prepare for the challenge in various types of universities or upper secondary schools, prepare students, teachers, pre-service teachers, or administrative staff.

The adaptation of a neuroanatomy course from f2f to online distance was described by D'Arcy et al. [14]. They paid an emphasis on image analysis. Most of the students reported positive affective outcomes for the course in project engagement and perceived mentorship received during the course. As expected, the students were frustrated by missing f2f communication and had problems in mastering image software skills, which caused a delay in producing consistent-quality data maps. To sum up, both useful approaches and areas for improvement were detected.

In higher education, students succeeded and failed in online distance instruction. For example, Zhang et al. [15] discovered that in the distance nursing course, learners

achieved lower test scores in self-efficacy, academic performance, and academic engagement compared to the group taught face-to-face.

Furthermore, Alam et al. [16] examined dental students' knowledge and perception of online teaching. They discovered that students preferred on-campus teaching to online teaching, as most students found online learning to be stressful and were quite unsatisfied. However, the difference in the mode of teaching did not affect their knowledge.

Compared to this approach, in Morocco, secondary school teachers used WhatsApp for ERT to ensure pedagogical continuity in education [17]. However, teachers also indicated a preference for both f2f and blended tutoring. Statistical analysis showed a significant correlation between the subjects taught f2f and distance education and a low correlation between teacher training and distance education.

Irrespective of the year of study [18], the students preferred in-person teaching and reported greater engagement, learning, and understanding during classroom teaching. More senior students, who had developed f2f contacts before the pandemic, found it easier to continue with interactions remotely. The first-year students were the most frustrated, but they considered it challenging to develop relationships remotely.

Relationship requirements also played an important role in the model using the process virtualization theory (PVT), as defined by Overby [19]. Alarabiat et al. [20] applied the model in the post-COVID era. They investigated the impact of the virtualization requirements of the learning process on students' satisfaction and their intention to continue using online learning. They developed a research model using PVT and validated it empirically with data obtained from an online questionnaire-based survey. The main results support the role of Sensory requirements. Relationship requirements have an impact on shaping students' satisfaction, which also has a significant influence on students' intention to continue online learning.

Last but not least, efficiency of learning is an important factor of ERT and ODI. In the field of medicine, a team of scholars at the University of Hong Kong [21] conducted a systematic review using the PRISMA statement when searching in the databases CINAHL, Cochrane, EMBASE, and Pubmed. In particular, they focused on the field of anatomy and surgical training and identified 182 non-duplicate studies. Besides other features, efficiency of learning was considered positive in most of them. However, the term is not clearly defined in the review.

In another study, the terms efficiency and effectiveness are not defined either [22]. However, the results from this study are expected to improve online distance learning by increasing these factors. A rather precise definition of the term efficiency was given by Gnawali et al. [23], Table 3, for science students in online higher education. Efficiency is defined by eight features, such as (1) this is the right time for the implementation of this program, (2) online content sharing and conference (Teams, Skype, Moodle, Google Classroom) tools are the best tools for the ODL, (3) there is an inappropriate timetable for online classes, (4) the ODL class schedule and time management of the class is very effective, (5) there was a lack of adequate technical support and ICT tools while training the initial phase at the campus, (6) there is a lack of time for practice at home, (7) the lack of time is the hindrance of your ODL mode class, (8) there is a lack of sufficient time to use ICT in the ODL classroom. However, this definition reflects more the features of online distance learning observed in the study than defines the efficiency of learning as, e.g., the relationship between measurement of achievement, such as a test score or the amount of time it takes a learner to correctly perform a task, and the perceived mental effort (PME) of the learner [24].

Regarding our research, we continuously monitored the course of online distance instruction in selected institutions in the Czech Republic during the COVID-19 pandemic restrictions (ERT, ODI) and post-COVID era.

In our research activities, we focused mainly on the range of tools exploited in ODI [25,26] and the advantages, disadvantages, and didactic recommendations for the next period of ODI [27]. Ours and other surveys conducted by the above-mentioned scholars

discovered gaps in the field and prepared conditions for carrying out the research described below.

## 2. Materials and Methods

### 2.1. Research Question and Objective

After months of teaching and learning under COVID-19 pandemic restrictions, the new era began. Despite the fact that numerous negatives and limits were discovered in ERT and ODI, there are also some potentials. Will the use of positively assessed ERT and ODI methods, approaches, strategies, tools disappear without a trace, or will we take their advantages in the new post-COVID era and enrich the f2f lessons enhanced by digital technologies, that is, blended learning, with them? The main objective of the research is to monitor students' opinions and discover whether the methods and tools used during the ODI can be found in BL one year later, thus enriching its range of tools.

### 2.2. Expectations

From a didactic point of view, we expect ODI to follow the main didactic principles, for example, to be open to everyone, to use illustrative study materials, appropriate to the learners' age and level of knowledge, to introduce the learning content step-by-step in a systematic but natural way, as they were designed and implemented in practice by Jan Amos Komensky (Comenius; 1592–1670) [28]. Regarding digital technologies in education, the intersection of technological, pedagogical, content knowledge is displayed in the TP(A)CK framework [29], and the objective of digital transformation of the learning content is shown through four levels of Substitution, Augmentation, Modification, and Redefinition of the SAMR model [30].

From the viewpoint of the content structure, four areas of ODI and BL were monitored. Within each area, expectations were set.

(1) First contact and communication. We expect that in ODI the first contact will be made at the very beginning of the semester because teachers do not want to waste time, while in BL, it will be made by the teacher f2f in the first lesson. During the semester, most of the communication will be held f2f during the lessons. In some cases, for example, sickness, teacher-student communication will be held online. It is usual that student-student communication is not conducted in the learning environment but through private channels, e.g., social networks.

(2) Learning content acquisition. Both in ODI and in BL, students should be motivated to learn. Most of the motivation is expected to be done f2f in BL. We expect that in ODI, learners will be advised how to study online, that is, how to communicate where to find and how to use study materials, presentations, and video recordings, how to work face-to-screen and ask and answer questions, discuss problems, do tests, and submit assignments, etc. We hope that, due to the fast development of digital technologies, new tools may appear in BL.

(3) Learning content delivery and assessment. In ODI, there is a preference to use one or two main channels to deliver the learning content. In BL, there is no reason to increase the number, as it confuses the students, and the f2f channel is also available. Logically, to assess students' performance, inform them about their progress and what they should focus on in further learning is done more frequently online in ODI and f2f in BL. We also anticipate that in BL, oral exams will be added to various types of online exams and tests and their combinations.

(4) Students' final feedback on online distance instruction. Using the four-level Likert scale, students express their agreement/disagreement with four statements: Teachers invested much effort in ODI, Students invested much effort in ODI, I appreciate ODI, and I did not learn much through ODI. We expect there will be supporters and opponents of ODI. However, their opinions will not be so sharp in Questionnaire 2 in autumn 2022 as

in Questionnaire 1 a year before. We hope students will consider ODI a helpful experience that they may utilize in the future.

### 2.3. Research Methods and Tools

Data were collected using two online questionnaires by the ex-post-facto method. Each questionnaire included 35 items that covered four fields of expectations. Twenty items required answers on the Likert scale, three items offered multiple-choice answers requiring one choice, and in eight questions, multiple-choice answers were allowed. Questionnaire 1 was piloted before by 20 randomly selected students of the upper secondary (US) school and higher education (HE) institution (10 + 10) and by six teachers working for both institutions and distributed in autumn 2021 to monitor the state during the final months of online distance instruction at selected educational institutions in the Czech Republic. Questionnaire 2 was piloted by another group of 10 students (5 + 5) and four teachers (2 + 2) and applied in autumn 2022 to monitor the state one year after the end of online distance instruction when blended learning might have been enriched by some methods and tools used in previous online distance instruction.

### 2.4. Research Sample

In total, 488 respondents participated in the research. Before starting the research, they were informed about the main objective and how the data collected from them will be treated. They voluntarily expressed their consent to participate in the research. Respondents were students of two types of institutions: upper secondary school, in particular, Emanuel Pötting Upper Secondary School for Medical Staff, Olomouc, and higher education institutions, in particular, Emanuel Pötting Vocational School for Higher Medical Staff, Olomouc, and Faculty of Education, Department of Information Technologies, University of Ostrava, Ostrava, Czech Republic. The structure of the research sample is displayed in Table 1.

**Table 1.** Research sample: The structure of respondents.

| Institution | Autumn 2021 | Autumn 2022 |
|---|---|---|
| Upper secondary: Total | 104 | 128 |
| Upper secondary: Male | 9 | 9 |
| Upper secondary: Female | 95 | 119 |
| Higher education: Total | 124 | 132 |
| Higher education Male | 59 | 33 |
| Higher education: Female | 65 | 99 |
| | 228 | 260 |
| Total: | 488 | |

Data were collected from students at upper secondary school and higher education institutions because some US learners might have enrolled to HE during the 18 months of pandemic restrictions. Therefore, both types of institutions were included in the sample. It was a convenience sample. Authors work with the institutions, two of them cooperated on the design of teacher training in ODI at US. Two authors lectured the teacher training courses at university.

## 3. Results

The results are structured according to the research content into four areas, displayed in figures, and described. Within each area, expectations are compared with reality, and potential interpretations are provided.

*3.1. The First Contact and Communication*

Generally, teaching and learning are based on contact and communication between the actors of the process. The sooner the first contact and the communication begins, the better it works for the whole process, no time is wasted. In ODI, almost all teachers made the first contact with students in the first or second week of the semester (98%), as we expected. However, in BL, they surpassed our expectations. Contrary to our expectation that the first contact would be made in the first f2f lesson, more than half of teachers contacted students a week before the semester started (57%) to confirm that there would be no ODI anymore and lessons would be held f2f in that semester, 36% of teachers made the first contact through technologies at the beginning of the semester.

In ODI, the results met our expectations about the regularity and frequency of communication: 72% of students think the communication was regular, 27% in ODI and BL describe the communication as irregular but conducted in needed (56% in BL). The frequency every week (25%) or less frequently (71%) in ODI mostly follows the school schedule. In BL, the frequency oscillates around 10%. It was also anticipated that after a long experience in ODI, teachers would use one channel/platform for communication. It was MS Teams during ODI (100%) while e-mail was not used at all. The main reason was that MS Teams was recommended by the Czech Ministry of Education as a common tool for Czech educational institutions of all levels [6]. Contrary to our expectations, in BL, teachers exploited two main tools for communication: MS Teams (83%) and e-mail (51%). This is a rather illogical and unexpected step if teachers, who are in a f2f contact with students, add another tool to MS Teams. The function of sending messages to a group is available in MS Teams, there is no need to use e-mail for this purpose. The service of sending individual messages is also available in MS Teams. However, writing answers to 51% of communication through e-mail messages seems to be highly demanding. The results are displayed in Figure 1.

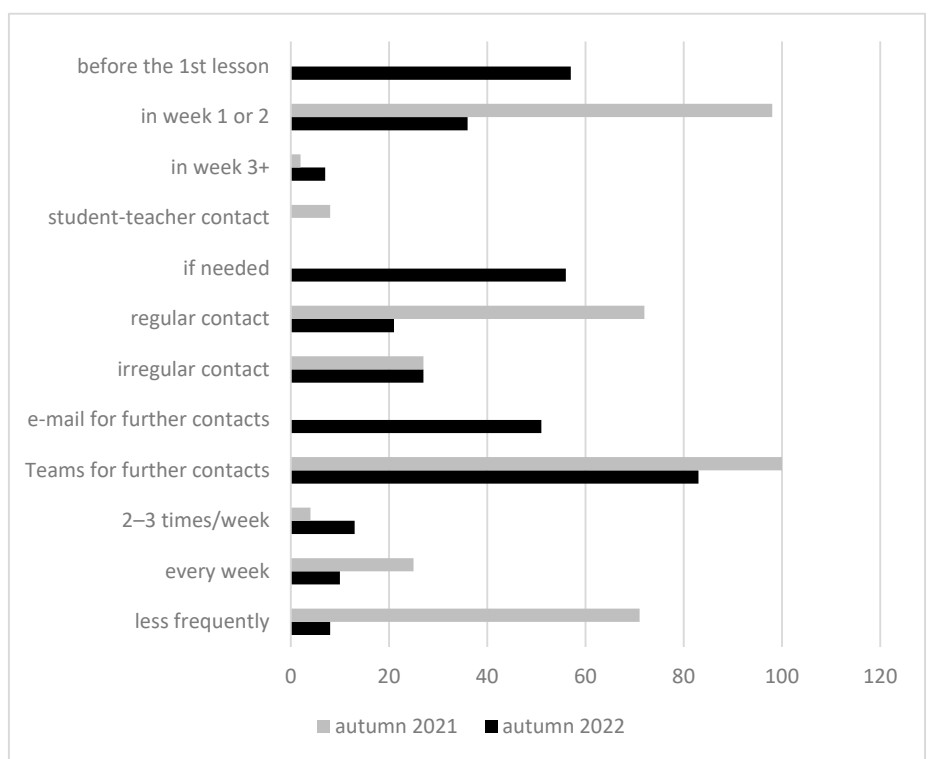

**Figure 1.** The first contact and communication.

### 3.2. Learning Content Acquisition

Motivation to learn is crucial for each activity. As expected, most students felt motivated by their teachers (81%) in BL, but only half of them were supported in ODI (44%). The support should have been much higher, we think, as it was in BL. In addition, students were instructed how to study online (65%) by digital study materials, that is, how to conduct teacher-student and student-student communication, how to study from digital texts, presentations, and video recordings, how to work in groups or teams, discuss problems, do tests, submit assignments, etc. We guess some of them could have felt motivated through this manual. As we expected, due to the f2f contacts in BL, the frequency of teacher-student communication decreased from 86% in ODI to 59% in BL. However, student-student communication increased from 28% to 35%. In addition, we can speculate about other communication through social media. In summary, it means a rather sharp increase in communication using digital technologies was detected in BL. Furthermore, a slight increase was found in BL (73%) compared to ODI (68%) in discussions on learning content (teacher initiated the discussion), in asking questions (teacher asked questions, students answered, 52% in BL, 11% in ODI), both activities were combined.

In the field of study materials, presentations were the format that was used by 96% of teachers in BL compared to 80% in ODI. An important increase was in other sources, from 37% in ODI to 81% in BL. In open answers, students specified, for example, TED talks—influential videos from expert speakers on education, business, science, technology, and creativity, they can choose whether to watch and listen with/without subtitles and transcript, which adjust understanding and follow-up work with the text to their level of knowledge. Furthermore, they mentioned various images, figures, diagrams, simulations, animations that help them acquire vocabulary, describe, and understand some processes in their future profession, etc.

Identical frequency of usage was detected in texts (58% in ODI; 57% in BL), a decrease was found in video recordings (61% in ODI; 49% in BL). While the frequency of using online exercises to practice new knowledge was also almost identical (70% in ODI; 67% in BL), a decrease appeared in using online tests with feedback (40% in ODI; 16% in BL), irrespective of the fact whether there was an explanation of a mistake or not (24% in ODI; 8% in BL). We hope that due to the fast development of digital technologies, new tools may appear in BL. The results are displayed in Figure 2.

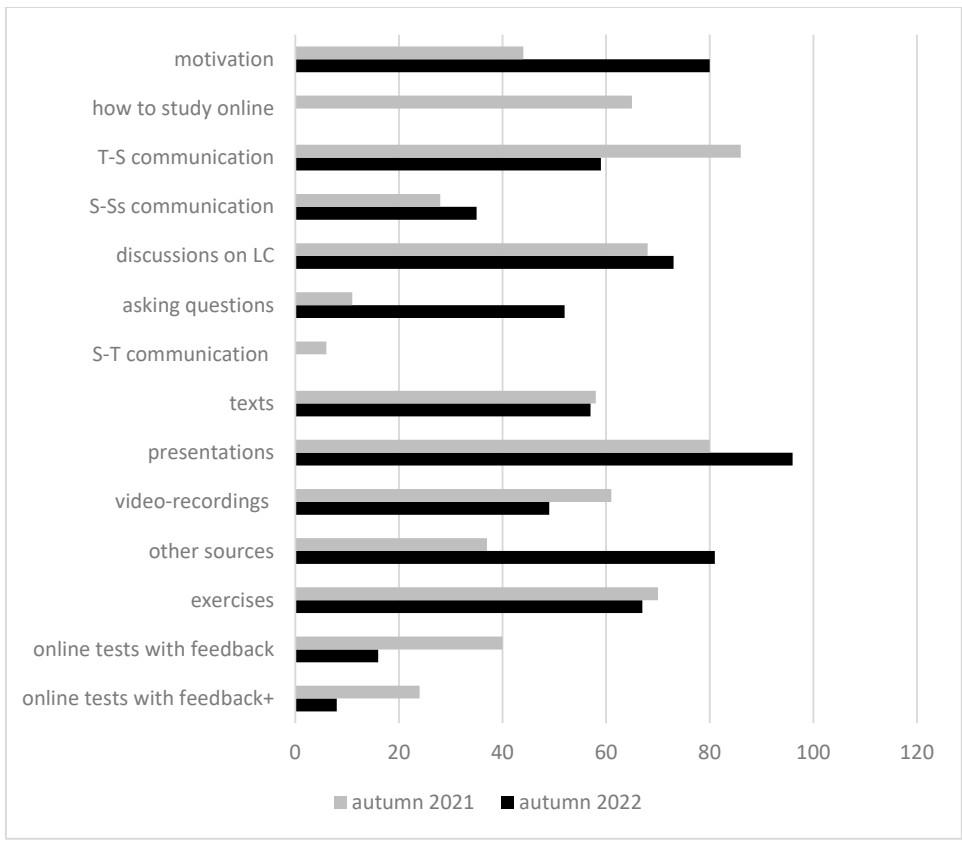

**Figure 2.** Learning content acquisition.

### 3.3. Learning Content Delivery and Assessment

In ODI, as we had expected, it was proved that one channel was used to deliver learning content, that is, study materials, exercises, online tests—100% of teachers exploited MS Teams for this purpose. In BL, MS Teams was involved slightly less (89%), but the learning content was also delivered through LMS Moodle (24%) and e-mail (34%), which broke the rule of one channel. Furthermore, regarding the assessment of students' knowledge, final exams were conducted not only online but also f2f. The exploitation of online oral exams (sometimes also called face-to-screen) decreased sharply from 45% in ODI to 5% in BL. Identically, the usage of online tests decreased from 55% in ODI to 8% in BL. Contrary to this, f2f oral exams were held by 55% of teachers while 68% of students sat for f2f printed tests in BL. These results reflected one or the other situation and followed our expectations. Surprisingly, students' progress was assessed more frequently in BL (by 33% of teachers) compared to 16% in ODI. Moreover, further learning development was directed to more students in BL (39%) than in ODI (23%). In addition, other methods of assessment were applied in ODI, for example, work on projects, seminar works, writing essays, etc. None of them was mentioned in BL. The results are displayed in Figure 3.

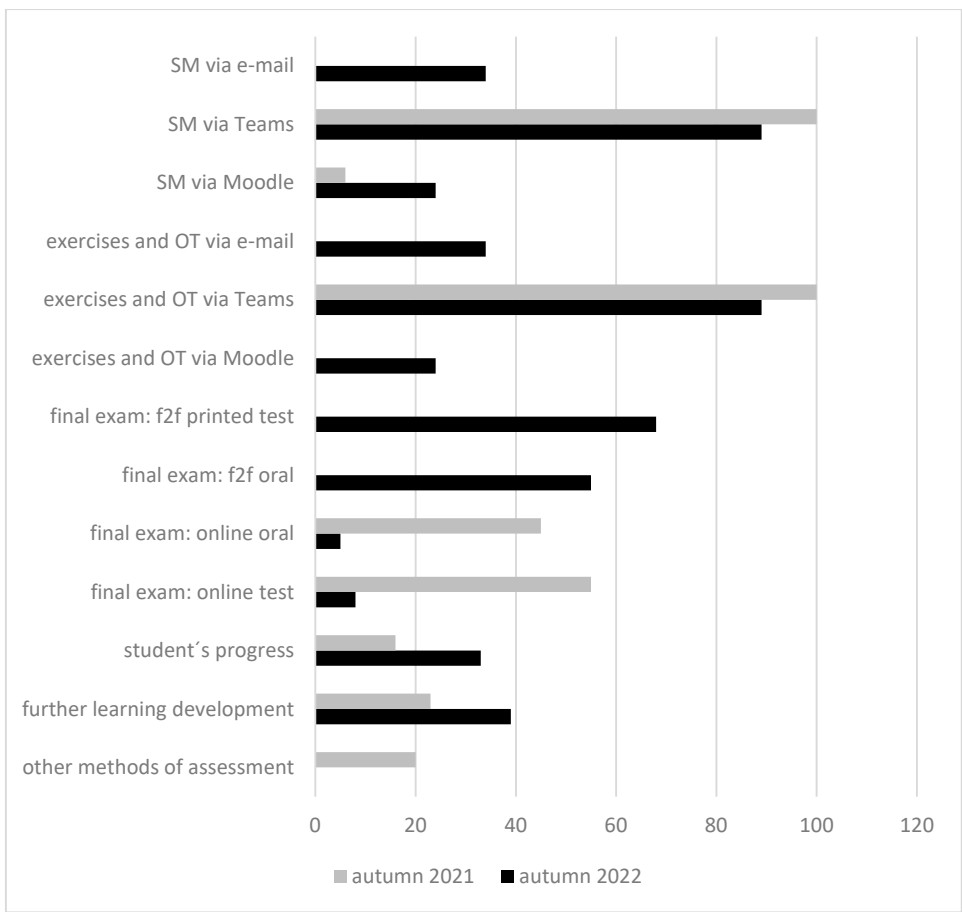

**Figure 3.** Learning content delivery and assessment.

### 3.4. Student's Final Feedback on Online Distance Instruction

Students' final feedback was collected in four parts. They were formed as statements, and students expressed the dis/agreement with them.

In statement 1 'Teacher invested much effort in ODI', students' negative assessment slightly increased in autumn 2022 (from 56% in autumn 2021 to 20% in autumn 2022 in fully agree, from 4% to 22% in rather disagree, contrary to the increase from 39% to 56% in rather agree).

In statement 2 'Students invested much effort in ODI', they were even more critical, there was a decrease from 37% in autumn 2021 to 19% in autumn 2022 in fully agree and from 50% to 38% in autumn 2022, contrary to the increase from 11% to 35% in rather disagree.

In statement 3 'I appreciate ODI', a sharp increase was detected in autumn 2022 (41%) compared to autumn 2021 (24%) in fully agree, a slight decrease in rather agree (from 41% to 38%), in rather agree, from 26% to 18%, and in rather disagree (from 26% to 18%). All these results give positive evaluation in favor of ODI one year after it was finished. Compared to this result, for example, 53% of 194 medical students preferred f2f teaching, but they accepted online lessons for teaching theoretical subjects. In the qualitative analysis, students clearly indicated that in subjects requiring teaching in a clinical setting or a patient exposure, student presence at the campus would be crucial to meet the learning objectives. In other subjects, online lessons are acceptable or even preferable because the students can watch the recorded lectures multiple times, which gives space to acquire the learning content. They agreed that interaction in online sessions is limited, and technical

problems and other challenges may appear during online examinations [31]. These problems may decrease students' satisfaction with ODI.

In statement 4 'I did not learn much through ODI', 60% of students rather disagreed in autumn 2021. Contrary to this, in autumn 2022, only half of them expressed the same opinion (27%), 36% rather agreed, and 24% fully agreed with the statement. When we consider statements 3 and 4, the question appears of whether there are some similarities between these two areas and what is hidden behind them—Why do the students appreciate ODI? Are the learning methods the reason, or how much they can learn, or is it the learning environment as a whole which may have been comfortable for some of them but stressful for others? Whatever the reasons are, our expectations were met—the data prove there are both supporters and opponents of ODI, but the number of those who appreciate this way of teaching and learning slightly increased from autumn 2021 to autumn 2021. While in June 2021, when ODI finished in the Czech Republic students, teachers, parents were tired of the 18-month-long period, often not caring about the learning result at that time. However, since the new school/academic year, gaps in knowledge were discovered step-by-step, followed by the need to fill them in. Thus, during the whole year, students were exposed to the missing knowledge. This requirement and effort that they had to make may have impacted their opinions on how much they learned through ODI. The results are displayed in Figure 4.

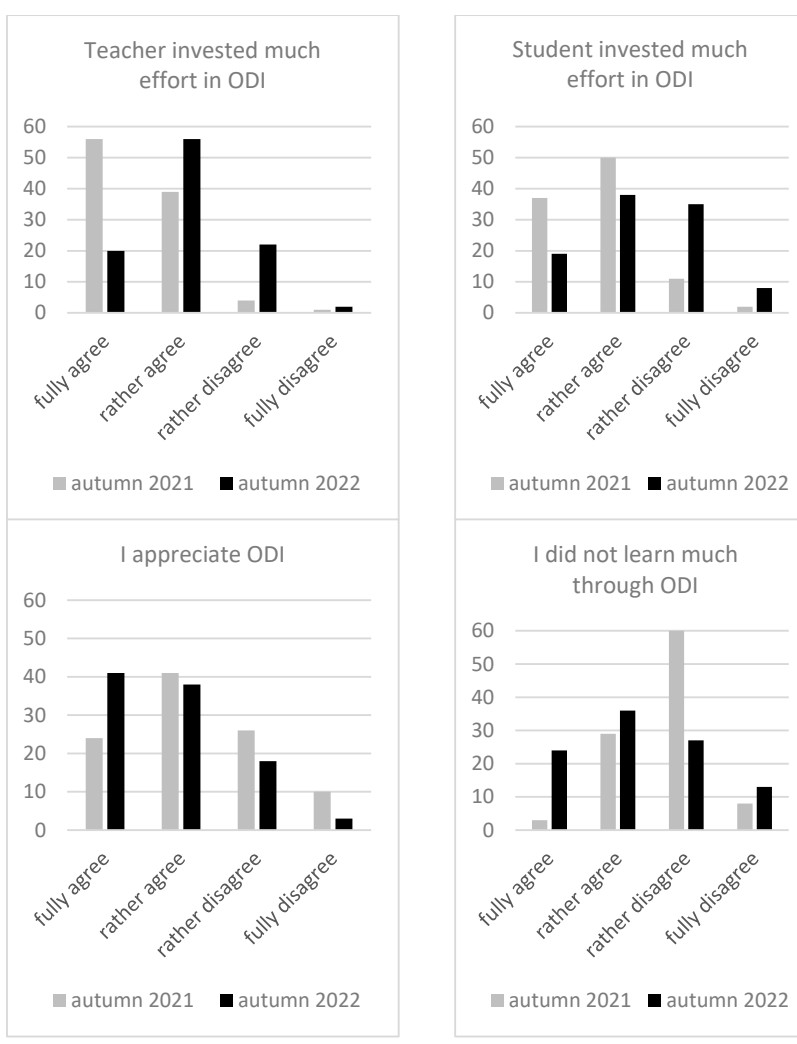

**Figure 4.** Student's final feedback on online distance instruction.

## 4. Discussion and Conclusions

Although we hope to overcome the times of COVID-19 restrictions in education, there is still not enough research carried out in the post-COVID period that pays attention to the implications of the process of transition back from online to f2f learning. In the context of our research results, we consider the work of Stoian et al. [32] to be important. They focus on identifying the aspects of online instruction preferred by students during the pandemic that students did not encounter in f2f lessons. Namely, students expressed their opinions on learning, teaching, assessment, and interaction with teachers and peers, with the aim to improve f2f education by contextually adapting it to their needs. This approach is rather similar to ours, including the use of an online questionnaire in the group of higher education students. They identified that 18.5% would appreciate study materials in an electronic format, an online course to support f2f lessons (15.1%), grid tests (26.4%). On the other hand, 38.6% of students did not find anything to borrow from online learning to improve f2f lessons, 47.1% did not find anything to borrow from online teaching, 11.9% of respondents did not discover any differences between the online and f2f assessment, and the f2f and online teacher-students interaction (23.6%). Switching from online to f2f lessons was very difficult (10%) or difficult (27.3%) for the same number of students as those who were indecisive (neither difficult nor easy 38.2%). These were the topics that we did not namely cover in our research, including the question of which form of instruction they consider more beneficial for the professional development (f2f was preferred by 38.7%, online learning 34.3% of respondents). Contrary to this result, a preferred form of education was online (53%, f2f 47%). However, practicing the skills was the reason why students consider f2f learning more beneficial for professional development. This conclusion is applicable to both technical university students in the Stoian et al. sample and (higher) medical staff and IT professionals in our research.

Another systematic review and meta-analysis in teaching medical (surgical) skills were carried out by Mao et al. [33], who aimed to assess the effectiveness of online video-based education compared with standard conventional lessons. In the majority of studies, no significant difference was found between the methods of conventional and video-based education in teaching basic surgical skills. Mao et al. conclude that basic surgical skills can be taught as effectively through online video-based lessons as conventional teaching methods, and online education should be used as an adjunct to medical curricula beyond the COVID-19 era.

Similarly to Simonova et al. [34], Rasli et al. [35] considered the COVID-19 pandemic an opportunity for educators and policymakers to rethink education systems. Their study focused on identifying strategies for higher education institutions when confronted with still unknown problems during and after the pandemic. They applied the Expert Opinion Method and involved a team of experts—five professors from the senior management of their universities. The experts proposed strategies for teaching during the COVID-19 period and set four major topics to be solved: (1) Resilience and Change Management, i.e., to develop a resilient recovery model to improve the ability to adapt to threats posed by the pandemic and reflect the need to rethink learning and make an educational paradigm shift from the pre-pandemic to post-pandemic state, (2) digital Transformation and Online Learning, when due to the need for ERT and ODI, US and HE institutions preparing professionals strive to remain competitive and provide high-quality education, (3) curriculum Change, once in process due to the pandemic, should focus on outcome-based learning and critical thinking as a minimum, and last but not least, (4) sustainability, which is expected to enhance transformative learning [36], and environmental and international networking. Based on 18 key excerpts from experts, the authors performed a thematic analysis, identified five dimensions, and integrated them into a framework. The framework included flexibility and agility, educational reform, digital transformation strategies through resilience, paradigm shift, and change management. A long-term target for higher education institutions is to ensure the sustainability of operations and performance (p. 8). As Rasli et al. conclude blended learning works as a sample of flexibility, not only

in higher education. In connection with our research, they had the post-COVID period in mind. It is much easier to deduce conclusions from the data collected ex-post-facto than to predict what may happen and prepare measurements not only to protect but even develop the system.

From the view of ODI evaluation, grading learners' performance should not be omitted. We can find the following features typical for this research: (1) Although the teachers aimed at simulating the online distance instruction compared to the "traditional" process to the maximum extent, in assessment, learners' effort to succeed in learning was also taken into account. In some cases, we can say that the assessment reflected more if the learners made an effort to achieve the learning objectives than whether they fully achieved them. In other words, the grades worked as one of the motivators more than an assessment factor. However, this did not work with test scores. (2) Types of tasks, homework, assessment were not the same as they had been in pre-COVID period, they reflected the restrictions. Therefore, presentations and online tests were frequently used, even in the courses where other types would have been more appropriate. (3) Not only theoretical knowledge that was mostly mediated through online distance lessons, but also practical activities were part of many courses. They were conducted face-to-face later on when the pandemic restrictions were canceled. Thus, the grades evaluating learners' knowledge after the course did not reflect the full state of the art. (4) Within the research, students attended more than 100 courses. Not all students enrolled in the course participated in the research. Learners' results in a selected course are displayed in Table 2.

**Table 2.** Learners' results in three periods in credit tests and exams in a selected course.

| Period of ODI | N | A | B | C | F | Mean Test Score (%) | Type of Assessment |
|---|---|---|---|---|---|---|---|
| Period 1 (06/20) | 20 | 3 | 2 | 13 | 2 | 72.7 | written online credit test |
| Period 1 (06/20) | 20 | 7 | 6 | 6 | 1 | - | oral f2f exam |
| Period 2 (01/21) | 19 | 12 | 3 | 4 | 1 | 82.6 | written online credit test |
| Period 3 (06/21) | 17 | 6 | 4 | 7 | 0 | - | oral f2f exam |

Online distance instruction: ODI; N; grades A: Excellent, B: Very good, C: Good; F: Failed; face-to-face: f2f.

As we can see, in period 2, after almost five months of true online distance instruction, students reached really high test scores that were 10% higher compared to the previous period. This was not an example of assessment working as a motivator from teacher's side but a result of learners' effort, student-student cooperation, and teacher-student discussions.

As a result of the COVID-19 pandemic in the field of education, the Czech government carried the program Together after COVID [37] that focused on two fields: (1) To improve learners' knowledge, i.e., to add the knowledge lacking due to the COVID-19 pandemic, (2) to support socializing, restoring social relations mainly among children, avoid risky behavior, support well-being, (3) to implement social and global topics in lessons, use methods that support an active approach to learners and make them active, (4) to provide teachers with methodological support to implement the topics correctly.

The development in the time of COVID-19 restrictions can also be seen from the viewpoint of selected Czech institutions that were under observation from the beginning of the pandemic. Based on the data collected from the respondents at these institutions, we were able to set the criteria that were used in this and previous surveys and adjust them as needed. In February 2021, after the first rather poorly coordinated period of teaching and learning under pandemic restrictions, most teachers and many students did not achieve the necessary level of digital literacy to be able to teach and learn successfully through digital technologies. Skype, e-mail, and social networks (or some of them) were known to teachers and students, but virtual platforms for educational purposes have not

been widely spread in the Czech Republic. Teachers were trained in a hurry, without state coordination and system [25]. In addition, some students had problems with autonomy in learning, which is a necessary prerequisite for successful online distance learning. They found numerous disadvantages in this way of instruction (an insufficient level of digital competency, a lack of technical equipment and support, problems in time management, etc.). On the other hand, other students were satisfied with online learning distance and detected various advantages (flexibility of learning, schedule, comfort of the home environment, etc.). As a result, didactic recommendations were set for the next period of ODI. It is not necessary to re-define the learning for the new era because the above-mentioned didactic principles (by Comenius), professional knowledge marked out by TP(A)CK Framework and SAMR model as a minimum are still 'valid' but rethinking of online distance teaching and learning is the topic of the day. Whether conducted f2f or online distance, enhanced by digital technologies or not, both teachers and learners aim to succeed in this process [27]. Their level of digital competency and knowledge of the latest technologies may help substantially in this process.

*Research Limits and Future Research*

Results are limited by the research sample. First, it is a convenience sample. Authors started cooperation with upper secondary and vocational institutions preparing medical staff several years before the COVID-19 pandemic. One of the authors (LF) has been working for them at a teaching position for two decades. Other authors (KK, IS) have been working in higher education for the same period. Second, the sample in not gender-balanced. However, the gender distribution follows a population distribution similar to upper secondary schools for medical staff and vocational schools for higher medical staff and university students in information technology study programs.

From the statistic point of view, in the Czech Republic, there are 91 upper secondary schools for medical staff and 43 higher education institutions preparing higher medical staff. The number of students slightly increases from year to year (in 2021/22: 16,314 students at upper secondary medical schools; 2020/21: 15,379; 2019/2020: 14,165) [38], even in the post-COVID times and despite the hard work and low salary in the field. On the other hand, there has been hardly any research in this field of education in the Czech Republic in the last decade, with the exception of Hubacek [39]. Therefore, among others, a good contribution of this research is that it opens the door to other surveys in the field.

In future research, emphasis should be paid to the outcomes of teacher training. The higher the level of teachers' digital skills, the better online distance didactics can be. Due to the fast technical and technological development, new applications are available. Students often use them for private purposes. However, if teachers are well trained, they can show them how they can also exploit the latest technology for learning. Thus, the contribution of technology is doubled using half the effort from the student side. Students feel familiar with the technology and they do not understand it as a constraint. Whether the process of individualization of learning is a step closer to a student is another research question.

**Author Contributions:** Conceptualization, I.S., L.F. and K.K.; Methodology, I.S., L.F. and K.K.; Validation, I.S. and K.K.; Investigation, I.S. and L.F.; Resources, L.F. and K.K.; Data curation, L.F. and K.K.; Writing—original draft, I.S. and L.F.; Writing—review & editing, I.S., L.F. and K.K.; Visualization, I.S. and K.K.; Supervision, K.K.; Project administration, L.F. All authors have read and agreed to the published version of the manuscript.

**Funding:** This research was funded by University of Ostrava, SGS 2023 project Didactic reflection of online distance instruction in the current face-to-face teaching and learning through the lens of students, teachers, and headmasters: best practices.

**Institutional Review Board Statement:** Not applicable.

**Informed Consent Statement:** Informed consent was obtained from all subjects involved in the study.

**Data Availability Statement:** Data are available on request from the corresponding author.

**Conflicts of Interest:** The authors declare no conflict of interest.

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
