# Peer review of "New Blended Learning Enriched after the COVID-19 Experience? Students’ Opinions"

_sustainability, doi:10.3390/su15065093_

Round 1

Reviewer 1 Report

The intensive process of the development of all social and educational spheres as a response to covid-19 restrictions in our society has accelerated the change of the status of the distance and blended learning ​​in the system of higher education, which influences not only the teaching methods, but also the completely new approaches in terms of the pedagogical object interaction. According to the modern requirements, one of the important factors is to absorb this or that information coming from various sources, and also to use it as a stable foundation for further self-improvement in the chosen activity. The general vector in the process of teaching ​​at the university becomes focused on the main areas of a particular specialty and involves a continuous search for the ways and approaches to educational process more efficiently.

Also the educational system faces the urgent task to form some capabilities for flexible changes in different ways and forms of pedagogical activities, and the developing of the key competencies that correspond to the main tasks of the students. I support the idea, that the most important quality of pedagogical thinking is flexibility. This is due to the specific teacher's tasks, namely: the need to switch from one activity to another, flexibility in using educational material for the upgrading of student’s personality. Blended learning seems to be a perfect way of studying due to mentioned circumstances.

However, no matter how positive and exciting the idea of ​​complete transforming the educational process into blended learning may seem to both teachers and students, it is important to remember that it should correspond to the educational content and goals of learning and provide the necessary results of the educational process.

The survey and the results of its research conducted by the authors seem quite convincing. However, in addition to the personal impressions of all participants of the educational process, its result is also important, namely: how did the overall results of students' educational success differ in the years of regular, distance and blended learning? What were the quantitative and qualitative coefficients of indicators of learning outcomes (grades received during the final exams) in all three periods? I believe that the research carried out by the authors will be quite complete and convincing if they add tables or charts of such comparisons to their article, and also draw conclusions based not only on the results of the survey, but also on the comparison of the above-mentioned coefficients.

Reviewer 2 Report

According to authors, the main objective of the presented research is to answer the question whether teachers enriched face-to-face lessons enhanced by digital technologies (i.e., blended learning) after the covid-19 pandemic with the methods and tools that they used during ODI in the pandemic. The results did not show much enrichment of blended learning using the experience from ODI.

The paper is well-written and structured. 

Although its an interesting study, it is very limited. It brings an analysis of a very restricted context, and it is not possible to see how face-to-face lessons enhanced by digital technologies (i.e., blended learning) can be efficiently enhanced in order to better support the learning process. Maybe a more deep analysis considering how to efficiently ennhace learning contents and how they impact in the learning process efficiency may bring interesting contributions to the community. In this context, including background and related work sections may bring more consistency to the paper.

Reviewer 3 Report

This work is of interest in the field of teaching. The results do not vary significantly in relation to the expectations that motivate them and are also in line with the perception we have in our university environment: exclusively online teaching has difficulties, face-to-face teaching brings some benefits and, above all, it can be seen that using one or the other with the best tools of each is preferable. However, it seems that the experience of the pandemic, both here and there, has not sufficiently left the use of online tools in face-to-face teaching.
I think that what is in the title, and what is highlighted in the paper, should be made clearer at the end of the abstract, instead of "We can speculate about their reasons".

Reviewer 4 Report

This research pretends to determine whether teachers improved face-to-face lessons with digital technologies following the COVID-19 pandemic using the methods and tools they used during online distance instruction. While presentations were more frequently used in blended learning than in ODI, teachers did not use a single channel to deliver study materials and communicate as they had done in ODI. This lack of consistency may have contributed to students believing it did not assist them in learning.

The article is not overly lengthy. It is simple to read, presenting the study and the stages of the process clearly and concisely without leaving out any essential details. The literature review demonstrates and frames a series of relevant studies to support the relevance and need to comprehend the challenges of the ongoing digital transformation and how mixed methods can be leveraged. While the findings are limited to the Czech Republic, they may shed light on the challenges and opportunities for blended learning in other countries.

The methodology used is sufficient. The method is robust due use of the two surveys, their suitability for the target audience, and validation through a pilot. Although statistical inference analysis was not performed, a more detailed characterisation of the sample, such as determining whether the number of responses represents the population, could have been performed. It should be clarified whether the surveys were adapted or self-authored. In future studies, the introduction of other factors, e.g., sociodemographic ones, may help to understand whether economic factors can influence this transformation.

Some typos fixes:

·        Line 27 introduces the acronym f2f instead of line 30.

·        Line 128 correct techers by teachers.

·        The acronyms US school and HE institution must appear on line 236 of the manuscript.

Congratulations to the authors for the research work carried out.

Round 2

Reviewer 2 Report

Authors improved the quality of the paper.